# Modeling the Effect of Annulus Fibrosus Stiffness on the Stressed State of a Vertebral L1 Body and Nucleus Pulposus

**DOI:** 10.3390/bioengineering11040305

**Published:** 2024-03-24

**Authors:** Oleg Ardatov, Jolita Pachaleva, Viktorija Aleksiuk, Algirdas Maknickas, Ilona Uzieliene, Raminta Vaiciuleviciute, Eiva Bernotiene

**Affiliations:** 1Faculty of Mechanics, Vilnius Gediminas Technical University, LT-10223 Vilnius, Lithuania; algirdas.maknickas@vilniustech.lt; 2Department of Regenerative Medicine, State Research Institute Centre for Innovative Medicine, LT-08410 Vilnius, Lithuania; jolita.pachaleva@imcentras.lt (J.P.); viktorija.aleksiuk@imcentras.lt (V.A.); ilona.uzieliene@imcentras.lt (I.U.); raminta.vaiciuleviciute@imcentras.lt (R.V.); eiva.bernotiene@imcentras.lt (E.B.); 3Faculty of Fundamental Sciences, Vilnius Gediminas Technical University, LT-10221 Vilnius, Lithuania

**Keywords:** annulus fibrosus, finite element method, hyperelasticity, nucleus pulposus, intervertebral disc, stiffness, vertebra

## Abstract

The investigation examines the transference of stiffness from intervertebral discs (IVDs) to the lumbar body of the L1 vertebra and the interactions among adjacent tissues. A computational model of the vertebra was developed, considering parameters such as cortical bone thickness, trabecular bone elasticity, and the nonlinear response of the nucleus pulposus to external loading. A nonlinear dynamic analysis was performed, revealing certain trends: a heightened stiffness of the annulus fibrosus correlates with a significant reduction in the vertebral body’s ability to withstand external loading. At a supplied displacement of 6 mm, the vertebra with a degenerative disc reached its yielding point, whereas the vertebrae with a healthy annulus fibrosus exhibited a strength capacity exceeding 20%. The obtained findings and proposed methodology are potentially useful for biomedical engineers and clinical specialists in evaluating the condition of the annulus fibrosus and predicting its influence on the bone components of the spinal system.

## 1. Introduction

Numerical studies can serve as an effective complement to the study of biomechanical processes occurring in the spinal system. The ability to parameterize models allows for the evaluation of the interactions between different objects with varying properties, while saving time and financial resources. This supplies the possibility of solving various problems, such as the transmission of stiffness from the intervertebral disc (IvD) to the vertebra. This task is particularly relevant in cases where the stiffness of the intervertebral disc changes, for example, in the case of soft tissue deterioration caused by degenerative diseases.

The IvD is an essential spinal unit that provides flexibility and supports compression, torsional and flexion loads [1]. IvD is consistently subjected to external forces and experiences different magnitudes and types of loading according to activity [2].

The IvD is composed of three main structures: the nucleus pulposus (NP), annulus fibrosus (AF), and two hyaline cartilaginous endplates, which connect the IvD with vertebral bodies [2]. NP is a gelatinous proteoglycan-rich structure surrounded by fibrocartilage AF. AF prevents the radial disc bulging of the NP by generating large hoop stress. NP tissue has a simpler and more homogeneous structure, whereas AF tissue is highly heterogeneous and differs from the inner to the outer region [3].

IvD is an avascular organ, and its mechanical behavior is crucial for cell nutrient transfer via diffusive mechanisms. Changes in the biomechanical properties of the IvD have a negative impact on flexibility, anatomy, and motion, and can result in the loss of normal IvD function and lead to the progression of IvD degeneration [1].

The subchondral bone of the vertebrae is also a significant contributor to various intervertebral disc diseases. Anatomically, the vertebra serves as the nearest communication part of the disc. In particular, the bone marrow of the vertebrae, which is highly cellularized and comprises a multitude of different immune and stromal cell types, presents numerous possibilities for responding to the factors released by the disc. The bone marrow serves as a vascularized gateway that facilitates communication between the IVD and the circulatory system [4]. When the disc undergoes degeneration, it impacts the mechanical load transmission to the vertebral body. Previous research discovered that the decompression of the lumbar IVD through enzyme injection or surgery can lead to the formation of lesions in the adjacent vertebrae within 24 weeks [5].

To conduct a thorough investigation, important aspects related to the properties of bone and soft tissue, as well as the geometry of the developed numerical models, must be taken into account. Recent studies offer various approaches to modeling the properties of bone and adjacent soft tissues. In recent studies, bone is most often represented as a linear elastic isotropic body [6,7,8]. Some authors take into account transverse isotropy [9], and studies can also be found in which bone tissue is modeled as a poroelastic body [10] or an elastoplastic continuum [11].

There is a much wider variety of proposed types of materials for the components of intervertebral discs. Some authors consider the annulus fibrosus to be a linear elastic material [12], while others consider it to be a hyperelastic matrix [13], viscoelastic matrix [14], or as having assigned poroelastic properties [15] in finite element models. The nucleus pulposus is generally modeled as a nonlinear material, and recent research offers various approaches using the following materials: an incompressible fluid-like material [16], hyperelastic neo-Hookean solid [13], hyperelastic Mooney–Rivlin solid [17], poroelastic solid [18], viscoelastic [14], and osmoviscoelastic solid [19].

The major drawback of those studies is the lack of specificity, namely, the absence of a comparison of elastic constants with the clinical condition of the patient. However, some investigators found a decrease in segmental stiffness with low-grade degeneration [20,21], while other studies observed a rise in stiffness with an increasing degeneration grade [22,23]. Comprehensive experimental studies of annulus fibrosus samples could shed light on changes in the mechanical characteristics of the annulus fibrosus in degenerative states. In our numerical investigation, we aim to emphasize the existence of this problem in this field, and using the results obtained from the numerical model, we strive to demonstrate the necessity of conducting physical experiments for a more detailed and thorough study of the contribution of annulus fibrosus to the shock absorbance and resistance to the mechanical overload in the spine.

Another important issue is the creation of a geometric model. The authors of [24] presented a finite element study of the lumbar vertebra; however, the influence of the intervertebral disc was not taken into account. Another study [25] proposed modeling the vertebral system by taking into account the interaction between the vertebral surface and the intervertebral discs, but the lumbar body was represented as a homogeneous continuum, and the influence of cortical and cancellous tissue was not considered. The authors of [26] studied mechanical characteristics using numerical models, but their studies did not evaluate the hyperelastic behavior of the intervertebral disc. The methods of modeling lumbar vertebrae with components such as cortical bone, cancellous bone, posterior elements, cartilage endplates, annulus fibrosus, and nucleus pulposus were presented in [27,28], but these studies aimed to study bone degeneration without considering the influence of the various stiffnesses of the intervertebral discs. 

In studies focused on modeling and analyzing the mechanical behavior of intervertebral discs, the primary emphasis is often placed on the nucleus pulposus, with the influence of the annulus fibrosus, beyond its role in retaining the nucleus pulposus, frequently marginalized or disregarded entirely. We propose the hypothesis that the stiffness of the annulus fibrosus itself governs the efficacy of the nucleus pulposus and, consequently, exerts a pivotal influence on the interaction between discs and vertebrae, including the efficient transmission of forces that mitigate bone tissue strain during external loading. In other words, this pertains to the problem of stiffness transfer from the disc to the vertebra in cases of annulus fibrosus degeneration.

In this study, we model the variation of an essential parameter of the annulus fibrosus, namely, its stiffness. While degenerative diseases do affect the entire disc, in our research, we aimed to focus on the annulus fibrosus. This approach allows us to elucidate its influence on the stressed state of the vertebral system and contribute to a deeper understanding of the specifical functioning of the annulus fibrosus. It also sheds light on its interactions with surrounding tissues, expressing the transfer of stiffness between softer and stiffer tissues. The proposed approach presented in this study, as well as the research results, may be useful for biomedical engineers and clinicians studying the mechanical interaction of tissues with different degrees of stiffness and stages of deterioration.

## 2. Materials and Methods

### 2.1. Geometry and Structure of the Model

A three-dimensional model of the lumbar vertebra L1 (Figure 1) was developed in several steps. To obtain the anatomical curvature of the lumbar body, DICOM images of a sixty-year-old woman suffering from third-degree arthritis were processed with 3D Slicer software (version 4.11) [29]. Based on the same images, the posterior elements were also exported as the STL format file, and then the whole model of lumbar vertebra was processed using MeshLab software (version 2021.05) [30]: the geometry was simplified (the initial number of mesh elements was reduced from 59,681 to 39,000) and smoothened (Laplacian smooth applied). The simplification result is shown in Figure 1a. Then, the file was exported to SolidWorks software (version 2023) [31], the final refinements were made using the ScanTo3D module, and the STL mesh was converted into a solid continuum body. In order to distinguish cortical and cancellous bone, a shell with a thickness of 0.5 mm was extruded in SolidWorks software. In addition, two horizontal surfaces mimicking endplates were created. They were given a slightly smaller thickness of 0.2 mm. These values correspond to the average values typical of lumbar bones [8]. The cancellous bone remained as a solid continuous body, and in order to account for its porous structure, a lower elastic modulus was assigned to it than to the cortical bone. The values of elastic constants are provided in Section 2.3.

Intervertebral discs (T12-L1 and L1-L2) were attached to the upper and lower surfaces of the endplates. Their height was set to 10 mm (Figure 1b). In our study, they consist of two parts: the outer annulus fibrosus and the inner nucleus pulposus. All components of the developed model can be seen in cross-section (Figure 1c). The most important geometrical parameters of the model are listed in Table 1.

Certainly, the presented model has several simplifications. Primarily, these include the absence of ligaments and the hyaluronan endplate. Additionally, trabecular tissue is modeled as a homogeneous continuum. Significant geometric simplifications also include perfectly flat horizontal surfaces of the vertebrae and the isotropy of the annulus fibrosus and nucleus pulposus. The absence of ligaments imposes some limitations on conducting numerical experiments—notably, it makes it inconsistent to study the model’s behavior under loading types such as bending or torsion. The absence of the hyaluronan endplate may also distort stress distribution on horizontal spinal surfaces; however, we assume that the compression test conducted on the model allowed us to identify the primary tissue interaction trends arising from the increased stiffness of the annulus fibrosus.

### 2.2. Problem Formulation

To assess the interaction of tissues with different mechanical behaviors, nonlinear elasticity theory was applied. Its selection is justified by the fact that bone tissue behaves predominantly linearly under load, while disc tissues (annulus fibrosus and nucleus pulposus) exhibit hyperelastic properties. The interaction between hard and soft tissues involves the mutual transfer of stiffness due to the transmission of external loads, and any change in the properties of any component of the spinal system can lead to changes in stress magnitude and redistribution. 

In dynamic analysis, the equilibrium equations of the system at time step *t* + ∆*t* are expressed as follows [31]:(1)[M]t+∆t{U″}(i)+[C]t+∆t{U′}(i)+Kt+∆ti∆Ut+∆ti={R}t+∆t−Ft+∆ti−1,
where [*M*] is the mass matrix, [*C*] is the damping matrix, *t* + ∆*t*[*K*](*i*) represents the stiffness matrix of the system, *t* + ∆*t*{*R*} represents the vector of externally applied nodal loads, *t* + ∆*t*{*F*}(*i* − 1) represents the vector of nodal forces at iteration (*i* − 1), *t* + ∆*t*[∆*U*](*i*) represents the vector of incremental nodal displacements at iteration (*i*), *t* + ∆*t*{*U*′}(*i*) represents the vector of total velocities at iteration (*i*), and [*M*]*t* + ∆*t*{*U*″}(*i*) denotes the vector of total accelerations at iteration (*i*), where the damping matrix [*C*] was neglected, [*C*] = 0.

Using the implicit time integration Newmark–Beta scheme and applying Newton’s iterative method, the above equations are expressed in the following form:(2)[K] t+∆t(i)  ∆U(i)={R} t+∆t(i),
where *t* + ∆*t*{*R*}(*i*) represents the effective load vector, and *t* + ∆*t*[*K*](*i*) denotes the effective stiffness matrix. The three-dimensional nonlinear problem was solved using SolidWorks software (Simulation module).

It should be noted that the applied theory imposes certain limitations. First and foremost, we consider only cases of instantaneous loading, meaning that this study does not address the consequences of loads that could manifest in the long term.

A specific difficulty lies in the verification of results. Primarily, this is due to the individual diversity of tissues and their properties, and the inability to compare the obtained results with a standard. However, we assume that it will be possible to capture the general trend of stiffness transfer changes in case of tissue degeneration.

Furthermore, it should be noted that nonlinear elasticity theory does not account for biological processes occurring in tissues. On the other hand, the components of the spinal system presented in the study are also objects of mechanics. Therefore, we consider this theory suitable for identifying the main patterns of changes in the stressed state of the model.

To reach this goal, the von Mises criterion was applied. It is defined in Equation (3):(3)$$\sigma_{y}=\sqrt{\frac{{\left(\sigma_{1}-\sigma_{2}\right)}^{2}+{\left(\sigma_{2}-\sigma_{3}\right)}^{2}+{\left(\sigma_{3}-\sigma_{1}\right)}^{2}}{2}},$$
where *σ*_1_, *σ*_2_, and *σ*_3_ are the maximum, intermediate, and minimum principal stresses, respectively, and *σ*_y_ is a yield stress.

### 2.3. Mechanical Properties of Model Components

The bone tissue was represented as an elastic–plastic isotropic continuum. The cortical tissue was assigned an elastic modulus of 8 GPa [7,8], while the elastic modulus of cancellous bone was set to 100 MPa [26,27]. Both components were assigned the same Poisson’s ratio of 0.3 [7,8]. To evaluate the strength of the vertebral body, the bone tissue was assigned a yield strength of 64 MPa [7,8].

To account for the nonlinear behavior of the intervertebral disc in case of compression, the nucleus pulposus was represented using the Mooney–Rivlin material model, with constants *C*_1_ and *C*_2_ set as 0.12 MPa and 0.03 MPa, respectively [17]. The Poisson’s ratio was assigned as 0.4995.

The annulus fibrosus is modeled as a linearly elastic continuum. In a review article [32], it is noted that the reported elastic constants typically fall within the range of 2–8 MPa (Young’s modulus), while the Poisson’s ratio is commonly around 0.45. In our study, we considered two cases. For the first case, we assumed a Young’s modulus of 2 MPa, and for the second case, it was 8 MPa. Additionally, we made the assumption that in the first case, we are examining a healthy annulus fibrosus, and in the second case, we are examining one affected by degenerative diseases. It should be noted that these assumptions are not absolute truths; however, they are based on known trends, such as the increase in tissue stiffness in degenerative diseases. Through numerical calculations and the interpretation of the results obtained, we attempted to verify the correctness of our assumptions.

### 2.4. Computational Model, Boundary Conditions, and Mesh

The loading scheme is presented in Figure 2a. Notably, the load is not directly applied to the investigated vertebra L1; instead, the model of the L1 vertebra along with two intervertebral discs (T12-L1 and L1-L2) was placed between two auxiliary vertebrae models (L2 and T12). It should be noted that the geometry of T12 and L2 models was developed on the basis of L1 surface curvature, so we should treat them as nominal models and cannot declare that their geometry is patient-specific. 

An additional intervertebral disc (L2-L3) was attached to the nominal L2 model, and a rigid constraint was applied to its lower surface. The nominal T12 vertebra model transfers the vertical displacement, which is applied to the upper surface of the T11-T12 disc. 

This loading scheme with the implementation of nominal models allowed for a more accurate transmission of stiffness from intervertebral disc T12-L1 to the lumbar body L1, and the calculated stresses on the L1 vertebra model were not distorted by either boundary conditions (rigid constraint causes stress concentrators) or the nearby location of the load application (Saint-Venant’s principle). 

The whole computational model was loaded until yield stresses (64 MPa) appeared on the L1 vertebra. Two tests were conducted with different properties of the previously obtained annulus fibrosus, described in the previous section. The maximum applied displacement value was 7.6 mm. 

To solve the equilibrium equations, the Intel Direct Sparse solver was used. To effectively adapt the finite element mesh to the complex curvature of the model, meshes were applied with volumetric (tetrahedral) finite elements (Figure 2b). Number of finite elements—647,055; number of nodes—118,902. The model is characterized by 352,964 degrees of freedom. 

## 3. Results and Discussion

The von Mises stress plot of the whole computational model is presented in Figure 3. As we can see, the maximum values of stress are concentrated on the cortical bone of the upper vertebra, closer to its connection with the intervertebral disc. It should be noted that this result is primarily due to the proximity of the applied load location (Saint-Venant’s principle), and considering the stresses obtained on the upper vertebra is not appropriate. The effect of Saint-Venant’s principle is clearly observed on the T11-T12 IvD of the computational model. As we can see, its upper surface is much more distorted than the others. Otherwise, the obtained plot of stresses looks plausible: high stress values are found on the bone components, and much lower values are found on the IvD. This effect can be explained by the different stiffnesses of the tissues. Additionally, as we can see from the presented figure, the stiffness of the annulus fibrosus does indeed affect the stress values on bone tissue. Figure 3a shows a model with a healthy annulus fibrosus, and yield was achieved at a displacement value of 6 mm, while for the same stress value, a model with degenerated annulus fibrosus required a significantly smaller displacement of 4.4 mm (Figure 3b).

Next, we will only consider the results obtained for the L1 vertebra and the adjacent intervertebral discs (T12-L1 and L1-L2). Figure 4 shows von Mises plots obtained at different displacement values for a model with an intervertebral disc with healthy annulus fibrosus. In Figure 4a, we observe a more uniform distribution of stresses over the entire surface of the cortical bone. It is colored green, and the mean value is around 17 MPa. At the points of maximum curvature, stress values slightly increase, reaching values of 33 MPa in some nodes, although they do not have significant characteristics. In this case, the safety factor is slightly less than fifty percent.

The von Mises stress plots for the model with a healthy annulus fibrosus and an applied displacement of 6 mm are shown in Figure 4b. As we can see, the stress distribution on the cortical tissue is no longer uniform: nodes with increased values are located in the narrowest section of curvature and reach up to 50 MPa. The yield limit was still not reached, but the safety factor significantly decreased, in this case to just over 20 percent.

Figure 4c shows the model with a healthy annulus fibrosus and a displacement of 7.6 mm. As we can see, the yield limit has been reached and nodes with critical stress values are formed on the front surface of the cortical tissue. However, this plot does not indicate that the intervertebral disc is no longer performing its function, but rather that the deformations caused by the external load exceeds the permissible limit for bone tissue.

Figure 4d–f represent a model with a degenerated annulus fibrosus. As we can see, even in the case of a displacement of 4.4 mm (Figure 4d), the stress distribution cannot be called uniform. The maximum stress value approaches 50 MPa, indicating that the intervertebral disc is no longer able to perform its function. Notably, under a similar load and with a healthy IvD, stress values were one and a half times lower. Moreover, a similar value of strength capacity was obtained at a greater displacement for a healthier disc (Figure 4b).

Figure 4e,f demonstrate exceeding critical stresses, and if in the case shown in Figure 4f, the main factor in reaching the yield stress is a large displacement (7.6 mm), then in the plot shown in Figure 4e, the negative effect of higher stiffness is vividly demonstrated: transferring stiffness from the disc to the vertebral body causes stresses that exceed the yield stress, and in such a model, failure is expected.

The von Mises stress plots for the model in cross-section are presented in Figure 5. Figure 5a,c reflect the stress distribution on the cancellous bone. As can be seen from the presented plots, the stresses on the cancellous bone are distributed relatively evenly, and their values are much lower than on the cortical shell. This observation is in good agreement with the regularities identified by [33], where during the overloading of lumbar vertebrae, the cortical shell initially takes the main load, and then, with further loading of the vertebra, internal forces are transferred to the cancellous bone. The mean stress value on the cancellous bone with healthy annulus fibrosus was 11 MPa (Figure 5a), whereas at the same displacement, the stress on the cancellous bone at degenerated annulus fibrosus was slightly higher—16 MPa (5c). In addition, with more deteriorated annulus fibrosus, higher stress values are based closer to the endplates.

The stress distribution on the vertebral body obtained in our study is similar to the results presented in [34]. In both cases, the most stressed area is the anterior wall of the cortical bone. In our study, the thickness of the cortical tissue is not a variable parameter; however, comparing the results of both studies provides grounds to assume that the stiffness of the annulus fibrosus will have an even greater impact on the strength of the vertebral body in conditions such as osteoporosis, and the consequent thinning of bone caused by it will significantly reduce the load-bearing capacity of the vertebral body.

Figure 5b,d show stress distributions for the inner side of the cortical shell. As in the case of cancellous tissue, the distributions look more uniform in the case of less deteriorated annulus fibrosus. Stress values on the inner side of the cortical shell do not exceed 40 MPa (Figure 5b), whereas in the case of degeneration, stress values reach up to 50 MPa (5d).

Von Mises stress plots for intervertebral discs are shown in Figure 6. As can be seen from the plots, the highest stresses occur on the horizontal surfaces of the intervertebral discs. However, it should be noted that the stressed layer is very thin and only covers one finite element deep into the disc, although it occupies the entire surface area. The obtained result can be explained by direct contact with the endplates. Therefore, it is advisable to consider stress distributions and their values throughout the volume of the intervertebral disc. For the intervertebral disc with healthy annulus fibrosus, the average stress value throughout the volume was 8 MPa (Figure 6a), whereas for degenerated annulus fibrosus, the average stress value increased by one-third: it is equal to 12 MPa (6b). Interestingly, on the stress plots presented in the cross-section, differences in stress distribution on the nucleus pulposus are not noticeable. For a more detailed study of their stress state, the nucleus pulposus is presented on separate plots (Figure 7). 

As can be seen from Figure 7, differences in stress distribution, as well as their values, are noticeable. The maximum stress for the nucleus pulposus in case of a healthy annulus fibrosus was 2 MPa (Figure 7a), whereas in the case of degenerated one it was 3 MPa (Figure 7b). However, it should be noted that this effect is due to direct contact, which is caused by the encirclement of the deteriorated annulus fibrosus. If we take the average stress values throughout the volume of the body, they are both equal to 1 MPa in both cases. 

For a more visual understanding of the obtained results, the stress values at different loads and degeneration stages are presented in a comparative diagram (Figure 8). As can be seen from the columns of the diagram, the cortical shell is strongly affected by the stiffness of the annulus fibrosus. In the case of healthy intervertebral disc, the strength capacity of the bone can vary from almost 50% (displacement 4.4 mm) to 20% (displacement 6 mm). From this, a recommendation can be formulated that with a diagnosed degeneration of the intervertebral disc, not only the joint surface of the vertebra should be evaluated, but also the overall condition of bone tissue. 

After performing calculations and analyzing results, we can summarize that considering the nonlinear properties of intervertebral discs is crucial to evaluating the strength properties, as it could play a significant role in the observed relationship between disc degeneration and the heightened risk of vertebral fractures. However, a major challenge remains the insufficient data on the mechanical behavior of tissues in certain diseases. Although general trends, such as tissue stiffness in degeneration, are known, specific values of necessary constants or parameters are either unavailable or contradictory.

While the benefits of numerical studies were highlighted, their full-scale implementation requires an experimental database. Regarding this study and its proposed methodology, improvements can be made in several areas. Firstly, the numerical model geometry can be enhanced by including crucial anatomical components, like ligaments for studying lumbar body processes in more detail, especially during complex loading cases such as flexion. Secondly, a more complex mathematical framework can be applied to incorporate properties such as viscoelasticity and damping, enabling research on the impact loads experienced by the vertebra during normal life cycles. To obtain more accurate results, advanced material models accounting for anisotropy in mechanical properties should be used, although this is difficult due to the unique nature of each bone and the lack of a template for accurate modeling. 

## 4. Conclusions

This study developed a numerical model to evaluate the stress state of hard and soft tissues in the lumbar spine. The conducted FEM calculations helped to identify the following significant trends:
In the case of a model with a healthy annulus fibrosus, yield was achieved at a displacement value of 6 mm, whereas for the same stress level, a model with a degenerated annulus fibrosus required a significantly smaller displacement of 4.4 mm.The mean stress value on the cancellous bone with healthy annulus fibrosus was 11 MPa, whereas at the same displacement (6 mm), the stress on the cancellous bone with a degenerated annulus fibrosus was higher (16 MPa). Additionally, with a more deteriorated annulus fibrosus, higher stress values were observed closer to the endplates.Noticeable differences in stress distribution, as well as their values, were observed. The maximum stress for the nucleus pulposus in the case of a healthy annulus fibrosus was 2 MPa, whereas in the case of a degenerated one, it was 3 MPa.In the case of a healthy intervertebral disc, the strength capacity of the bone can vary from almost 50% (at a displacement of 4.4 mm) to 20% (at a displacement of 6 mm). From this, a recommendation can be formulated: with the diagnosed degeneration of the intervertebral disc, not only should the joint surface of the vertebra be evaluated, but also the overall condition of the bone tissue. This can be particularly important in cases of osteoporosis, where the thickness of the cortical shell can be reduced.


The proposed modelling approach can be useful for biomedical engineers and clinicians; however, for a more comprehensive implementation of the model and the proposed method in medical practice, extensive experimental studies of biological samples are necessary. For this reason, our further research in this area will focus on studying the mechanical properties of intervertebral disc tissue samples. This will allow us to supplement the numerical model, thus making calculations more reliable.

## Figures and Tables

**Figure 1 bioengineering-11-00305-f001:**
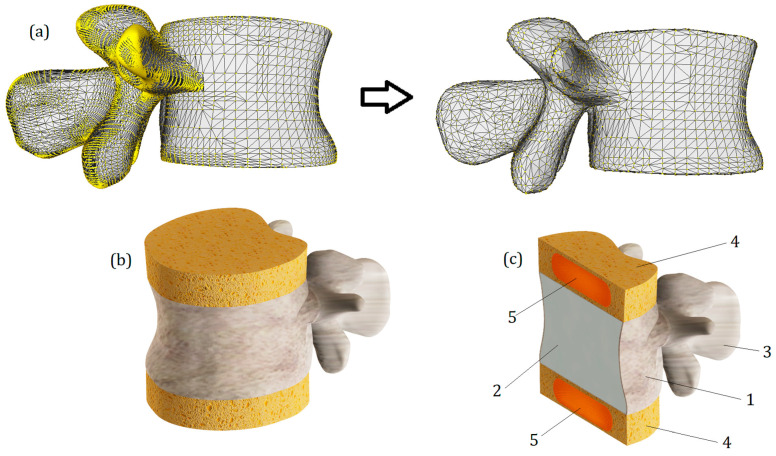
(**a**)—The simplification and smoothening of the model with MeshLab. (**b**)—Final rendered numerical model of the vertebra L1 with IvDs. (**c**)—Section view of the model: 1—cortical bone; 2—cancellous bone; 3—posterior elements, 4—annulus fibrosus; 5—nucleus pulposus.

**Figure 2 bioengineering-11-00305-f002:**
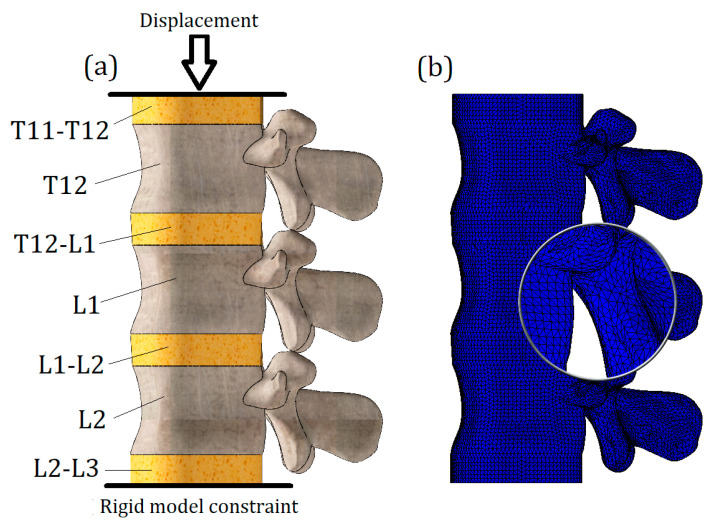
(**a**): Schematization of load; (**b**): finite element model.

**Figure 3 bioengineering-11-00305-f003:**
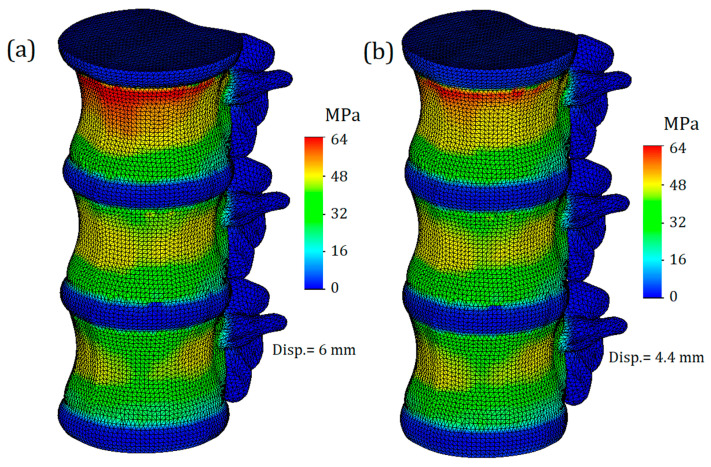
The trimetric von Mises stress plot of the whole computational model: (**a**)—with healthy annulus fibrosus; (**b**)—with degenerated annulus fibrosus.

**Figure 4 bioengineering-11-00305-f004:**
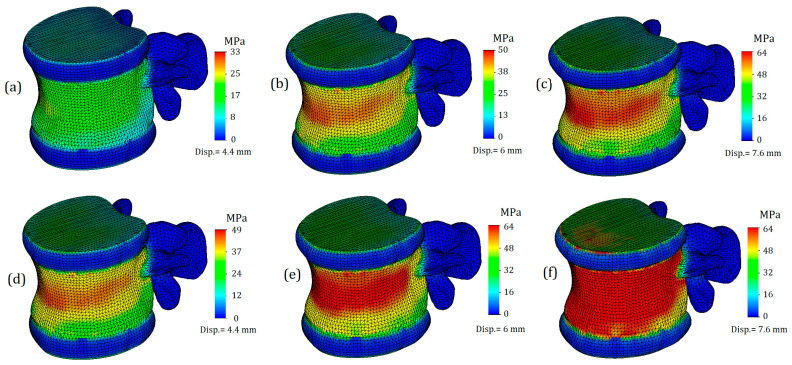
Von Mises stress plots on L1 vertebra in case of various displacement values: (**a**–**c**)—healthy annulus fibrosus; (**d**–**f**)—degenerated annulus fibrosus.

**Figure 5 bioengineering-11-00305-f005:**
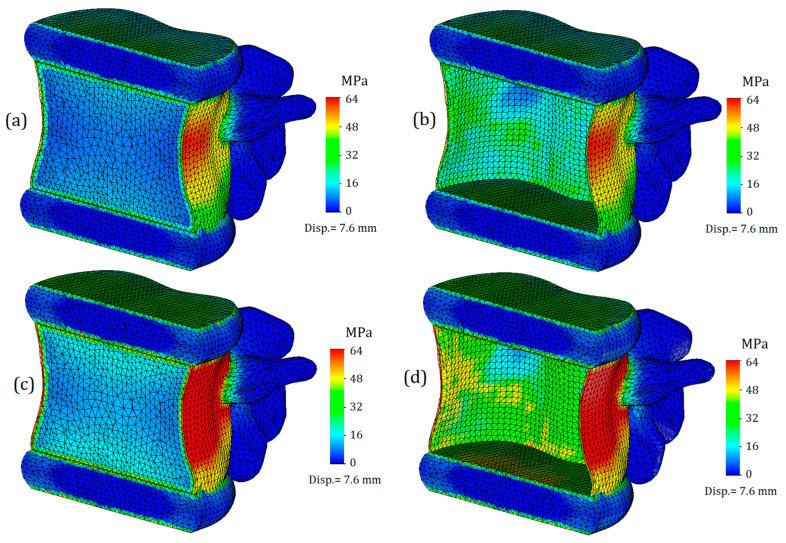
Von Mises stress L1 section view in case of 7.6 mm displacement: (**a**,**b**)—healthy annulus fibrosus; (**c**,**d**)— degenerated annulus fibrosus.

**Figure 6 bioengineering-11-00305-f006:**
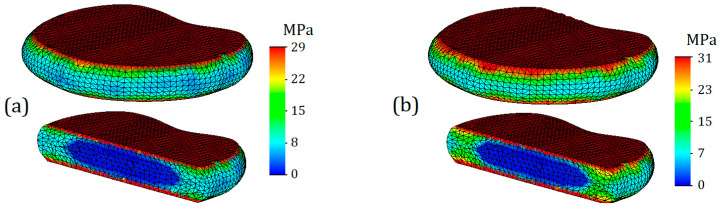
Von Mises stress plots on annulus fibrosus (T12-L1) in case of 7.6 mm displacement: (**a**)—healthy annulus fibrosus; (**b**)—degenerated annulus fibrosus.

**Figure 7 bioengineering-11-00305-f007:**
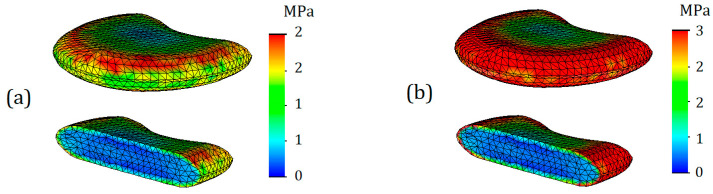
Von Mises stress plots on nucleus pulposus (T12-L1) in case of 7.6 mm displacement: (**a**)— healthy annulus fibrosus; (**b**)—degenerated annulus fibrosus.

**Figure 8 bioengineering-11-00305-f008:**
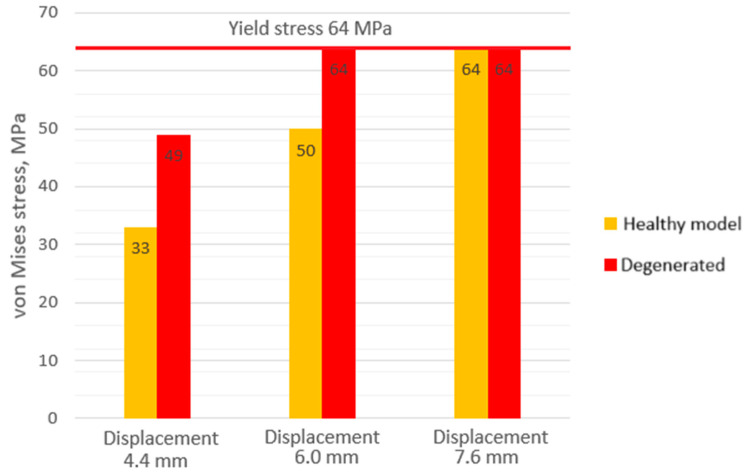
Maximum von Mises stress values on cortical shell of the lumbar body.

**Table 1 bioengineering-11-00305-t001:** Principal geometrical parameters of the model (undeformed).

Parameter	Value
Height of the vertebra, mm	30
Height of the IvD, mm	10
Area of the endplate, mm^2^	935
Volume of the annulus fibrosus, mm^3^	6076
Volume of the nucleus pulposus, mm^3^	2497
Total volume of the IvD, mm^3^	8573

## Data Availability

The data presented in this study are available on request from the corresponding author.

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
