# Peer review of "Modeling the Effect of Annulus Fibrosus Stiffness on the Stressed State of a Vertebral L1 Body and Nucleus Pulposus"

_bioengineering, 2024, doi:10.3390/bioengineering11040305_

Round 1

Reviewer 1 Report

Comments and Suggestions for Authors

 Comments and Suggestions for Authors

The complicated and unique structure of the intervertebral disc IVD tends to support the weight of the entire body in addition to a wide variety of loadings and dynamic motions on the spinal cord. To date, the researchers have paid great attention to exploring various modeling of lumbar IVD mechanics under different physical conditions, to better elucidate the functions of the native disc tissues under various loading conditions. The manuscript entitled “Modeling the Effect of Annulus Fibrosus Stiffness on Stressed 2 State of Vertebral L1 Body and Nucleus Pulposus” has also a similar objective but lacks extensive investigation. Although the manuscript has provided alluring findings it needs further elaborations for the final recommendation for the publication.

1.    The abstract is written well but please precisely mention and add details like this calculation is associated with either static or nonlinear dynamic calculations of von Mises stresses so that the obtained results are good guidance for the development of patient-specific vertebra reliability.

2.    In the material and methods section a table could be added enlisting the general physical parameters of the vertebral body. This effort could enhance the reader’s understanding. The table should enlist the general features of vertebral body like height, the cross-section area of IVD , cross-section area of the nucleuaa pulposus , fiber direction etc…

3.    In addition a second table representing the unit types and material properties of the segment used in FEmdel shall also be a bonus.

4.    In line 182 several mechanical properties are reported please precisely explain weather it is under which type of mechanical loading either compressive or in tension.

5.    Please also discuss more elaborately the impact of thickness in the findings presented for Maximum von Mises stress values on the cortical shell of the lumbar body. For instance, it is reported that In the case of a cortical shell with a thickness of 0.4 mm, maximal stress is about 17 MPa due to the minimal BV/TV ratio and the maximum external load (0.75 MPa), reaching 43% of the yield stress. Ref. https://doi.org/10.3390/app9153013

6.    Please add some future works to the manuscript.

7.    The conclusion must be enumerated as per the order of the findings in the paper like key results revealing the analysis of the stress distribution and also highlighting the maximum and minimum thickness in number as mentioned in L347.

Comments on the Quality of English Language

Moderate editing of the English language required

Author Response

Response to Reviewer 1 Comments

Dear Reviewer,

Thank you very much for taking the time to review this manuscript. Please find the detailed responses and list of corrections and improvements below.

Point 1:. The abstract is written well but please precisely mention and add details like this calculation is associated with either static or nonlinear dynamic calculations of von Mises stresses so that the obtained results are good guidance for the development of patient-specific vertebra reliability.

Response 1: The abstract is improved and the information on nonlinear dynamic analysis is added.

Point 2: In the material and methods section a table could be added enlisting the general physical parameters of the vertebral body. This effort could enhance the reader’s understanding. The table should enlist the general features of vertebral body like height, the cross-section area of IVD , cross-section area of the nucleus pulposus, fiber direction etc…

Response 2: Thank You, we’ve added the table with information about geometrical parameters of the model. The information about fiber direction is added to model limitations (in our work the IvD is considered isotropic).

Point 3: In addition a second table representing the unit types and material properties of the segment used in FEmdel shall also be a bonus.

Response 3: We’ve added the information about finite elements, used in our calculations are volumetric, and in order to fit the complexed curvature of vertebral body, the finite elements were tetrahedral.

Point 4: In line 182 several mechanical properties are reported please precisely explain whether it is under which type of mechanical loading either compressive or in tension.

Response 4: Thank You, we’ve clarified, that elastic constants are used for compression.

Point 5:. Please also discuss more elaborately the impact of thickness in the findings presented for Maximum von Mises stress values on the cortical shell of the lumbar body. For instance, it is reported that In the case of a cortical shell with a thickness of 0.4 mm, maximal stress is about 17 MPa due to the minimal BV/TV ratio and the maximum external load (0.75 MPa), reaching 43% of the yield stress. Ref. https://doi.org/10.3390/app9153013

Response 5: Thank you, we mentioned the provided article in the Discussion section and used the results given in it to compare with this work. This made it possible to emphasize the importance of the thickness of the cortical layer in the conclusions also.

Point 6: Please add some future works to the manuscript.

Response 6: Thank you, we have added some thoughts on the further development of this work in conclusions.

Point 7: The conclusion must be enumerated as per the order of the findings in the paper like key results revealing the analysis of the stress distribution and also highlighting the maximum and minimum thickness in number as mentioned in L347.

Response 7: The conclusions are now improved, the most important findings are enumerated.

Comments on the Quality of English Language: Moderate editing of the English language required

We’ve performed the Proofreading, the English in paper is now improved.

Reviewer 2 Report

Comments and Suggestions for Authors

-      Line 81 – “In our theoretical work..” this is actually beside being theoretical, also work where numerical simulation has been made.

-      Line 111 – “It will also shed light on its interactions with surrounding tissues.” – Could you point this out a little bit more?

Author Response

Dear Reviewer,

Thank you very much for reviewing our manuscript. Please find the the list of improvements below.

Point 1:. Line 81 – “In our theoretical work..” this is actually beside being theoretical, also work where numerical simulation has been made.

Response 1: Thank You very much, we have made amends to this sentence.

Point 2: Line 111 – “It will also shed light on its interactions with surrounding tissues.” – Could you point this out a little bit more?

Response 2: Thank you, we have added argumentation to this statement and explained it in more detail.

Round 2

Reviewer 1 Report

Comments and Suggestions for Authors

Dear Authors.

Thank you for making the corrections. I recommend manuscript for the potential publication.

Comments on the Quality of English Language

Minor editing of English language required

Author Response

Dear Reviewer,

We are thankful for your suggestions. The English has now been improved